# Comparative study on cytogenetics and transcriptome between diploid and autotetraploid rice hybrids harboring double neutral genes

Lin Chen[1,2,3©], Haibin Guo[2,4©], Shuling Chen[1,2,3], Huijing Yang[1,2,3], Fozia Ghouri[1,2,3], Muhammad Qasim Shahid [ID][1,2,3]*

**1** State Key Laboratory for Conservation and Utilization of Subtropical Agro-Bioresources, South China Agricultural University, Guangzhou, China, **2** Guangdong Provincial Key Laboratory of Plant Molecular Breeding, South China Agricultural University, Guangzhou, China, **3** College of Agriculture, South China Agricultural University, Guangzhou, China, **4** Center of Experimental Teaching for Common Basic Courses, South China Agricultural University, Guangzhou, China

© These authors contributed equally to this work.
* qasim@scau.edu.cn

**Data Availability Statement:** All relevant data are within the paper and its Supporting Information files. Transcriptome data have been deposited to

## Abstract

Double pollen fertility neutral genes, $Sa^n$ and $Sb^n$, can control pollen sterility in intersubspecific (*indica × japonica*) rice hybrids, which has excellent potential to increase rice yield. Previous studies showed that polyploidy could increase the interaction of three pollen sterility loci, i.e. *Sa*, *Sb* and *Sc*, which cause pollen sterility in autotetraploid rice hybrids, and hybrid fertility could be improved by double neutral genes, $Sa^n$ and $Sb^n$, in autotetraploid rice hybrids. We compared cytological and transcriptome data between autotetraploid and diploid rice hybrid during meiosis and single microspore stages to understand the molecular mechanism of neutral genes for overcoming pollen sterility in autotetraploid rice hybrids, which harbored double neutral genes. Cytological results revealed that the double neutral genes resulted in higher pollen fertility (76.74%) and lower chromosomal abnormalities in autotetraploid hybrid than in parents during metaphase I, metaphase II, anaphase I and anaphase II. Moreover, autotetraploid rice hybrid displayed stronger heterosis than a diploid hybrid. Compared with diploid rice hybrid, a total of 904 and 68 differently expressed genes (DEGs) were identified explicitly in autotetraploid hybrid at meiosis and single microspore stages, respectively. Of these, 133 and 41 genes were detected in higher-parent dominance and transgressive up-regulation dominance, respectively, which were considered autotetraploid potential heterosis genes, including a meiosis-related gene (*Os01g0917500*, *MSP1*) and two meiosis specific-genes (*Os07g0624900* and *Os04g0208600*). Gene Ontology (GO) and Kyoto Encyclopedia of Gene and Genomes pathway (KEGG) analysis revealed that DEGs significantly enriched in amino acid metabolism and photosynthesis metabolism. These results indicated that meiosis-specific and meiosis-related genes, and amino acids and photosynthesis metabolism-related genes contribute to higher yield and pollen fertility in autotetraploid rice hybrid. This study provides a theoretical basis for

NCBI SRA database (SRA accession: PRJNA656120).

**Funding:** This work was supported by the NSFC to MQS (31850410472), the Natural Science Foundation of Guangdong Province of China to HBG (2018A030310212) and the Opening Foundation from the State Key Laboratory for Conservation and Utilization of Subtropical Agro-Bioresources to HBG (SKLCUSA-b201402). The funders had no role in study design, data collection and analysis, decision to publish, or preparation of the manuscript.

**Competing interests:** The authors have declared that no competing interests exist.

**Abbreviations:** qRT-PCR, Quantitative real-time polymerase chain reaction; GO, Gene ontology; KEGG, Kyoto Encyclopedia of Genes and Genomes; DEGs, Differentially expressed genes; DEGs$_{HP}$, Differentially expressed genes between hybrid and parents; DEGs$_{PP}$, Differentially expressed genes between parents.

molecular mechanisms of heterosis in autotetraploid rice harboring double neutral genes for pollen fertility.

## Introduction

Rice is one of the most key cereal crops which feeds more than half of the world's population and is essential to food security. Hybrid rice has been recognized as one of the most suitable and sufficient technologies for increasing rice production. Asian cultivated rice mainly differentiated into *indica* and *japonica* after long domestication [1]. At present, *indica-japonica* hybrid rice has shown more substantial yield potential. Still, partial sterility has been a significant barrier to the utilization of the strong heterosis present in intersubspecific hybrids. One of the critical factors is the development of aborted pollen grains. So far, more than 50 loci controlling *indica* ×*japonica* sterility have been identified [2, 3]. Pollen fertility of *indica-japonica* hybrids is controlled by at least six loci (*Sa*, *Sb*, *Sc*, *Sd*, *Se* and *Sf*), which showed that it is a complex phenomenon. Gene interactions at pollen sterility loci could produce partially sterile pollens. Neutral alleles for pollen fertility ($S^n$) that do not interact with typical *indica* ($S^i$) and *japonica* ($S^j$) alleles provide a platform to exploit the strong hybrid vigor derived from intersubspecific crosses by controlling the reproductive obstacles between *japonica* and *indica* hybrids. Substantial evidence has shown that neutral genes, $S_a^n$, $S_b^n$, $S_c^n$, $S_d^n$, and $S_e^n$, provide valuable gene resources to surmount the pollen sterility related to the respective locus [4–8].

Autotetraploid rice is another viable option for increasing the yield of rice. Autotetraploid rice has a double genome compared with diploid rice and has wide adaptability [9–12]. Intersubspecific autotetraploid rice hybrids have more significant biological and yield potential than diploid rice [13–16]. Our previous research had demonstrated that polyploidy increased chromosomal abnormalities and enhanced pollen sterility loci interactions that cause low pollen fertility in autotetraploid hybrids [17–21]. However, the autotetraploid rice hybrids, which harbored double-neutral genes, $S_a^n$ and $S_b^n$, displayed high seed setting, high pollen fertility (>70%) and significant positive hybrid vigor for yield and yield-associated traits [22–24]. Cytological observations also showed a higher frequency of bivalents and normal chromosome behaviors in these hybrids than parents during meiosis. The double neutral genes, $S_a^n$ and $S_b^n$, could overcome hybrid sterility in diploid and autotetraploid rice hybrids [7, 24].

In recent years, the transcriptome analysis has become particularly widespread. The rapid development of next-generation sequencing has opened exciting avenues for investigating gene expression and function. The RNA-seq has been employed to identify differentially expressed genes between autotetraploid and diploid rice hybrids that heterozygous ($S^iS^j$) at *Sa*, *Sb* and *Sc* pollen sterility locus during pollen development. The results revealed that the polyploidy enhanced epistatic interactions between alleles of these three sterility loci, and the interaction at *Sb* was more robust than other loci [20, 25]. T449 is an important autotetraploid rice line, which contains double neutral genes, $S_a^n$ and $S_b^n$. The genomic variants were detected by re-sequencing between diploid (E249) and autotetraploid rice line (T449), and polyploidy not only induced abrupt changes in the expression patterns of meiosis-related genes which resulted in the abnormal chromosome behavior but also caused variations in the expression levels of saccharide-related genes [23]. High pollen fertility was found in the hybrids developed by crossing T449 and near-isogenic lines. Several hybrid combinations were generated by crossing T449 with neo-tetraploid rice lines, and differentially expressed genes were detected between parents and hybrids in nine tissues at different development stages. The results showed significantly higher expression patterns of necessary starch synthase and saccharides

metabolism-related genes. Many meiosis-related and specific genes were found to be up-regulated in the hybrid compared to the parent with low seed set [23].

In this study, we developed autotetraploid rice hybrids by crossing T449 with E24-4x, which contained double neutral genes, $S_a^n$ and $S_b^n$, to evaluate the effect caused by the polyploidy, and its diploid couterpart was used as control. Moreover, chromosome behavior during PMC meiosis and morphological traits were investigated in the hybrids and parents with different ploidy levels. Transcriptome analysis of anther development was also executed during a single microspore stage and meiosis to understand the gene expression profiles in the autotetraploid rice hybrid. The functional analysis of these transcripts may offer valuable information to decipher the molecular mechanism underlying heterosis by using neutral genes for pollen sterility loci in autotetraploid rice hybrids.

## Materials and methods

### Plant material

A diploid rice cultivar, E249, and its autotetraploid rice line, T449, harboring $Sa^n$ and $Sb^n$ double neutral genes for pollen sterility loci, were used as maternal lines, and crossed with E24 and E24-4x to develop diploid rice hybrid (hereafter referred as $DF_1$) and autotetraploid rice hybrid (hereafter referred as $AF_1$). E249 (DN18) harbored $Sa^n$ and $Sb^n$ double neutral genes for pollen sterility loci, and E24 is a near-isogenic line of Taichung 65, which has the same genetic background as Taichung65, except at $Sa$, $Sb$ and $Sc$ pollen sterility loci. T449 and E24-4x were developed by our research group from diploid rice E249 and E24 through colchicine-mediated chromosome doubling and self-crossed for more than 25 generations, respectively. These materials were grown at the research station of South China Agricultural University (SCAU) under natural environment, and management practices were done according to the recommendations of the area.

### Cytological investigations

The chromosome behaviors and configuration were investigated according to Chen *et al*. [23]. In short, the inflorescences of parental lines and $F_1$ were fixed in Carnoy's solution (ethanol: acetic acid = 3:1) for 24 h, and then kept at 4°C in 70% ethanol. After the removal of the floret, the anther was retained in a small drop of 1% acetocarmine on a microscope slide. After 2–3 min, a coverslip was used to cover the microscope slide and observed under a Motic BA200 microscope. The pollen fertility was detected by staining with 1% $I_2$-KI under a microscope (Motic BA200) [26].

### Evaluation of heterosis and agronomic traits

The mid parent heterosis (MPH) and high parent heterosis (HPH) were evaluated by the following formula: MPH = ($F_1$− MP)/MP × 100%, and HPH = ($F_1$− HP)/HP × 100%, where MP represents the mean value of two parents, HP represents the value of best parent, and $F_1$ indicates the performance of hybrid [12]. Agronomic traits, including filled grains per plant, 1000-grain weight, total grains per plant, effective number of panicles per plant, plant height, grain length and width, grain yield per plant, and seed set were investigated as described previously [12].

### RNA-seq and data analysis

All the tissues of parents and hybrids were collected in three biological replicates and stored at -80°C for RNA isolation. The total RNA was isolated according to the manufacture's protocol

of the TRIzol Reagent (Life Technologies, California, USA). The library was prepared according to the manual instructions as described previously [23, 24]. The low-quality data, including reads containing sequencing primer and nucleotides with quality score lower than 20, and sequencing adaptors, were discarded. All the data has been submitted to NCBI (Accession IDs: PRJNA656120, and transcriptome data of T449 and E249 was downloaded from NCBI under the accession number PRJNA436888 [23].

Differentially expressed genes in different samples were identified by Venny software (http://bioinfogp.cnb.csic.es/tools/venny/index.html). Cluster 3.0 software was used for hierarchical clustering of all genes after normalization. For functional categorization, gene ontology (GO) analysis was conducted by using the agriGO v2.0 (http://systemsbiology.cau.edu.cn/agriGOv2/). Transcription factor (TF) analysis was performed based on transcription factor data [27].

## Mapping of differentially expressed gene (DEG) to rice QTLs

The $DEGs_{HP}$ (DEGs between the parents and hybrid are labelled as $DEGs_{HP}$) were mapped onto 26 yield-related traits and 1019 yield-related QTLs using gene coordinates from the MSU Rice Genome Annotation Project. Rice QTLs with physical locations on the MSU Rice Genome Annotation Project were downloaded from Gramene [28].

## qRT-PCR validation

To validate the data of RNA-Seq by qRT-PCR, 10 DEGs were arbitrarily selected. Primer Premier 5.0 software was used to design gene-specific primers, and specific primers were examined in the NCBI database (S1 Table in S2 File). We take total RNA from sequenced samples, and the first-strand cDNA was produced using the Transcriptor cDNA Synthesis Kit (Roche) according to the manual instructions. The qRT-PCR reaction was performed on the Lightcycler480 system (Roche), and PCR profile was 30 s at 95˚C, with 40 cycles of 95˚C denaturation for 10 s and 60˚C annealing and elongation for 30 s. All qRT-PCR reactions were executed in three biological replications. The relative expression patterns of genes were estimated using the $2^{-\Delta\Delta Ct}$ method [29].

## Results

### Pollen fertility and chromosome behavior of hybrids and parents with different ploidy levels

In this study, T449 and E249, harboring $S_a{}^n$ and $S_b{}^n$ double neutral genes for pollen sterility loci, were used to develop autotetraploid rice hybrid (hereafter referred as $AF_1$) and diploid rice hybrid (hereafter referred as $DF_1$) by crossing with E24-4x and E24, respectively. There were different alleles at three pollen sterility loci (i.e. *Sa*, *Sb*, and *Sc*) in diploid and autotetraploid parents, so these hybrids presented allelic "interactions" at three pollen sterility loci. In diploid hybrid, pollen fertility was 68.67%, which was lower than its parents, and mid parent heterosis (MPH) and high parent heterosis (HPH) values for pollen fertility were -24.17% and -24.98%, respectively. However, pollen fertility was 76.74%, which was similar to their parents, and MPH and HPH values for pollen fertility were 4.43% and 1.59% in autotetraploid rice, respectively (Table 1; Fig 1). We detected high pollen fertility (>65%) in different ploidy hybrids with double neutral genes at *Sa* and *Sb* pollen sterility loci, and these results showed that different ploidy hybrids had no interaction at *Sa* and *Sb* loci. In addition, the MPH and HPH of autotetraploid hybrid for pollen fertility were higher than diploid hybrid, which exhibited that the effect of double neutral genes was stronger in autotetraploid hybrid.

**Table 1. Pollen fertility of hybrids and parents with different ploidy levels harboring genetic interactions at *Sa*, *Sb* and *Sc* loci.**

| Material | Genotype at *Sa*, *Sb* and *Sc* pollen sterility loci | Pollen fertility (%) | MPH | HPH |
|---|---|---|---|---|
| E249 | *nn/nn/jj* | 90.58±1.79 | -24.17 | -24.98 |
| DF₁ | *ni/ni/ij* | 69.44±2.88 | | |
| E24 | *ii/ii/ii* | 92.56±1.22 | | |
| T449 | *nnnn/nnnn/jjjj* | 75.03±2.26 | 4.43 | 1.59 |
| AF₁ | *nnii/nnii/iijj* | 76.22±3.82 | | |
| E24-4x | *iiii/iiii/iiii* | 70.94±0.97 | | |

E24 and E249 indicate diploid rice lines, and DF₁ indicates their F₁ hybrid; E24-4x and T449 indicate autotetraploid rice lines, and AF₁ indicates their F₁ hybrid.

Carmine acetate staining was employed to investigate the chromosome behavior in hybrids and parents with different ploidy (S1 and S2 Figs in S1 File). The chromosome behavior of diploid rice hybrid and its parents was similar, but in autotetraploid rice, the chromosomal abnormalities of AF1 hybrid was lower than its parents in the metaphase I, metaphase II, anaphase I and anaphase II (Fig 2; S2 Table in S2 File). These results indicated that autotetraploid hybrid chromosome behavior was better than parents.

## The phenotype of parents and hybrids with different ploidy levels

The plant height (PH) was significantly higher in hybrids than parents (S3 Table in S2 File). For the effective number of panicles per plant (EP), 1000-grain weight (TGW), filled grains per plant (FG), grain yield per plant (GYP), seed set (SS), MPH and HPH were higher in autotetraploid hybrid than a diploid hybrid (S3 Table in S2 File; S3 Fig in S1 File). In particular, MPH and HPH for filled grains per plant (FG) were 149.22% and 62.21%, for grain yield per plant (GYP) were 196.68% and 97.71%, and for seed set (SS) were 214.68% and 158.33%, respectively. These results showed that heterosis was more substantial in autotetraploid hybrid than diploid counterpart.

## Transcriptome profiles of rice anthers

To understand the gene expression profile in the autotetraploid rice hybrid, the RNA-seq experiments were conducted with different ploidy hybrids and their parents in anthers at

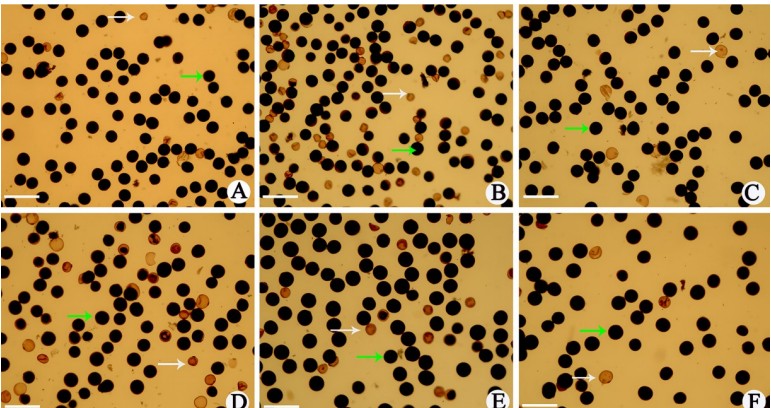

**Fig 1. Pollen fertility of hybrids and parents with different ploidy levels.** A, E249 (Diploid parent harboring double neutral genes); B, DF₁ (F₁ diploid hybrid); C, E24 (Diploid Parent); D, T449 (Autotetraploid parent harboring double neutral genes); E, AF₁ (F₁ Autotetraploid hybrid); F, E24-4x (Autotetraploid parent). Round and fully stained pollens represent fertile pollens (green arrows), while other represent abnormal pollens (white arrows). Bar = 100μm.

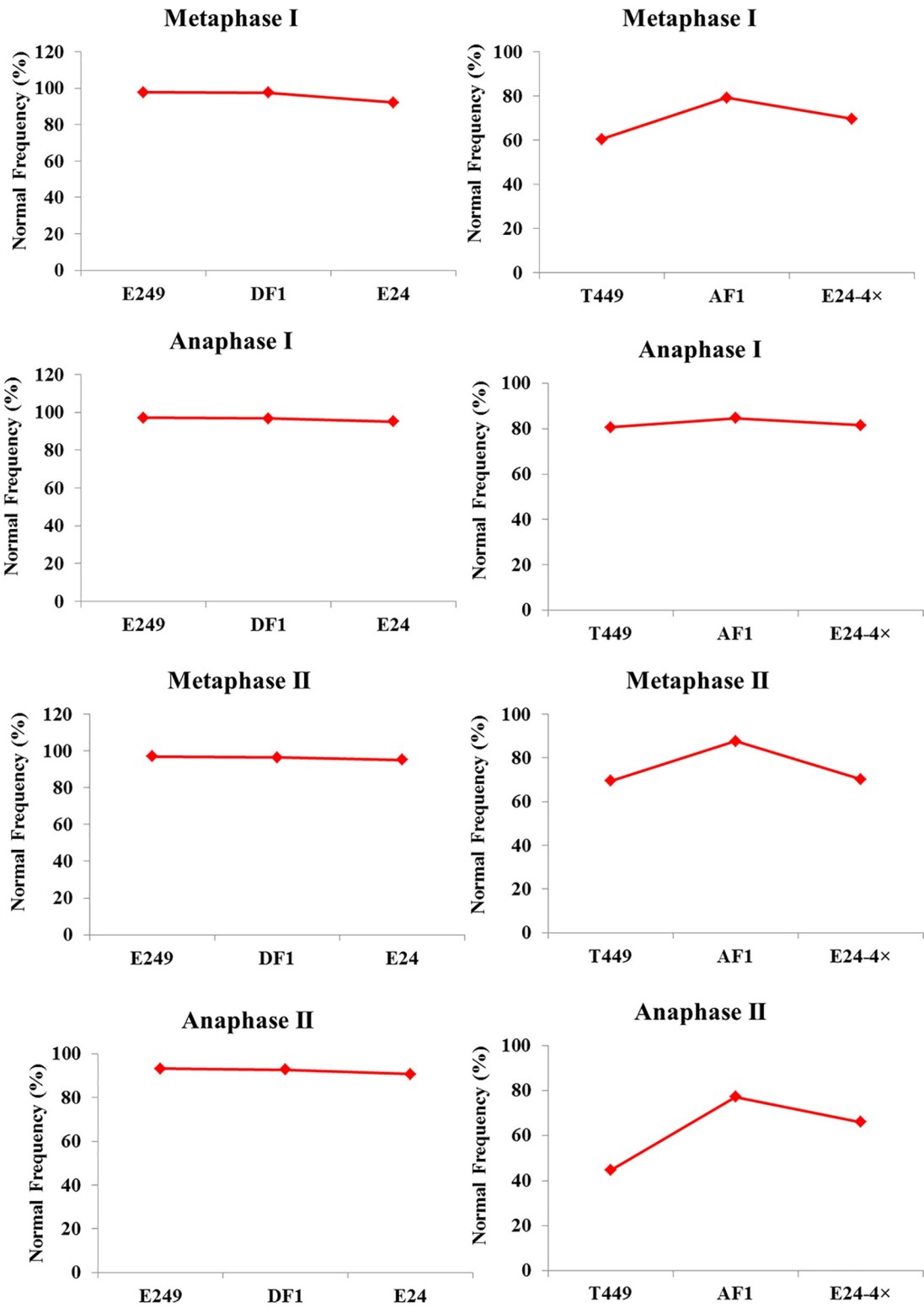

**Fig 2. Frequency of normal cells in different ploidy hybrids during PMCs meiosis.**

meiotic and single microspore stages. At meiosis and single microspore stages, the raw reads of the 36 samples were ranged from 18 to 22 million clean reads in anthers (S4 Table in S2 File). The clean reads were mapped onto the Nipponbare reference genome, and 84.28% to 90.41% annotated transcripts of the reference genome was detected in our material (S4 Table in S2 File). We removed some of the weak correlation coefficient samples, and kept more than 0.8 correlation coefficient (S4 Fig in S1 File). In total, ten genes were randomly selected from RNA-seq data to validate by qRT-PCR, including MADS gene (MADS56, *Os10g0536100*), meiosis related-gene (*Os03g0800200*), F-box gene (*Os06g0713400*), 60S acidic ribosomal protein (*Os12g0133050*), glutathione S-transferase (*Os01g0933900*), signal peptidase homologue (*Os05g0297900*), two hypothetical proteins (*Os12g0550600* and *Os12g0550600*) and two expressed proteins (*Os03g0249700* and *Os06g0574900*). The qRT-PCR results were similar to the RNA-seq data (S5 Fig in S1 File), which demonstrated that the RNA-seq data is dependable.

Differentially expressed genes (DEGs) were identified by using the two filter conditions (i.e. fold change (FC) higher than or equal to 2 and false discovery rate (FDR) less than or equal to 0.05). Using these standards, we identified 168 to 6640 DEGs in autotetraploid and diploid rice at the meiosis stage, and 7449 and 1480 DEGs were identified in diploid and autotetraploid rice, respectively (Table 2). DEGs between the parents and hybrid are labelled as DEGs$_{HP}$, and the DEGs between the parents are called as DEGs$_{PP}$. In total, 3604 and 1696 DEGs$_{HP}$ were found in diploid and autotetraploid rice at the meiosis stage, and 5345 and 549 DEGs$_{HP}$ at the single microspore stage, respectively (Table 2). A total of 904 and 68 DEGs$_{HP}$ were specifically identified in autotetraploid hybrid at meiosis and single microspore stages (Table 2). We primarily concentrated on these DEGs$_{HP}$ to detect genes related to autotetraploid fertility and heterosis.

The GO analysis of 904 DEGs$_{HP}$ showed that 41 GO terms were significantly enriched at the meiosis stage. In the biological processes category, 33 GO terms, including cellular nitrogen compound metabolic process, carboxylic acid biosynthetic process, DNA-dependent and regulation of transcription, were significantly enriched. In molecular function, eight GO terms, such as electron carrier activity, transcription factor activity and iron ion binding, were significantly enriched (Fig 3). No significant GO term was detected in the single microspore stage.

KEGG pathway analysis showed that 137 of 904 DEGs were associated with 21 functional terms at the meiosis stage, and 7 of 68 DEGs were enriched in 5 functional terms at the single microspore stage (Fig 4). Those, as mentioned earlier, 137 and 7 DEGs were enriched in 75 and 8 subcategories, and 16 and 2 significant terms were detected at the meiosis and single microspore stage, respectively (Table 3).

## Association of DEGs$_{HP}$ with QTLs for yield-related traits

We mapped DEGs$_{HP}$ detected in autotetraploid hybrid, which were specifically expressed in autotetraploid at meiosis, to identify QTLs associated with grain yield in the rice genome

**Table 2. Number and classification of DEGs identified in diploid and autotetraploid rice.**

| Development stage | Sample | DEGs$_{PP}$ | DEGs$_{HP}$ | | DEGs$_{HP}$ |
|---|---|---|---|---|---|
| | | | M/F$_1$ | F/F$_1$ | |
| Meiosis stage | diploid | 3562 | 2842 | 1145 | 3604 |
| | autotetraploid | 2711 | 1529 | 256 | 1696 |
| | autotetraploid-specific | | | | 904 |
| Single microspore stage | diploid | 6640 | 859 | 4848 | 5345 |
| | autotetraploid | 1436 | 395 | 168 | 549 |
| | autotetraploid-specific | | | | 68 |

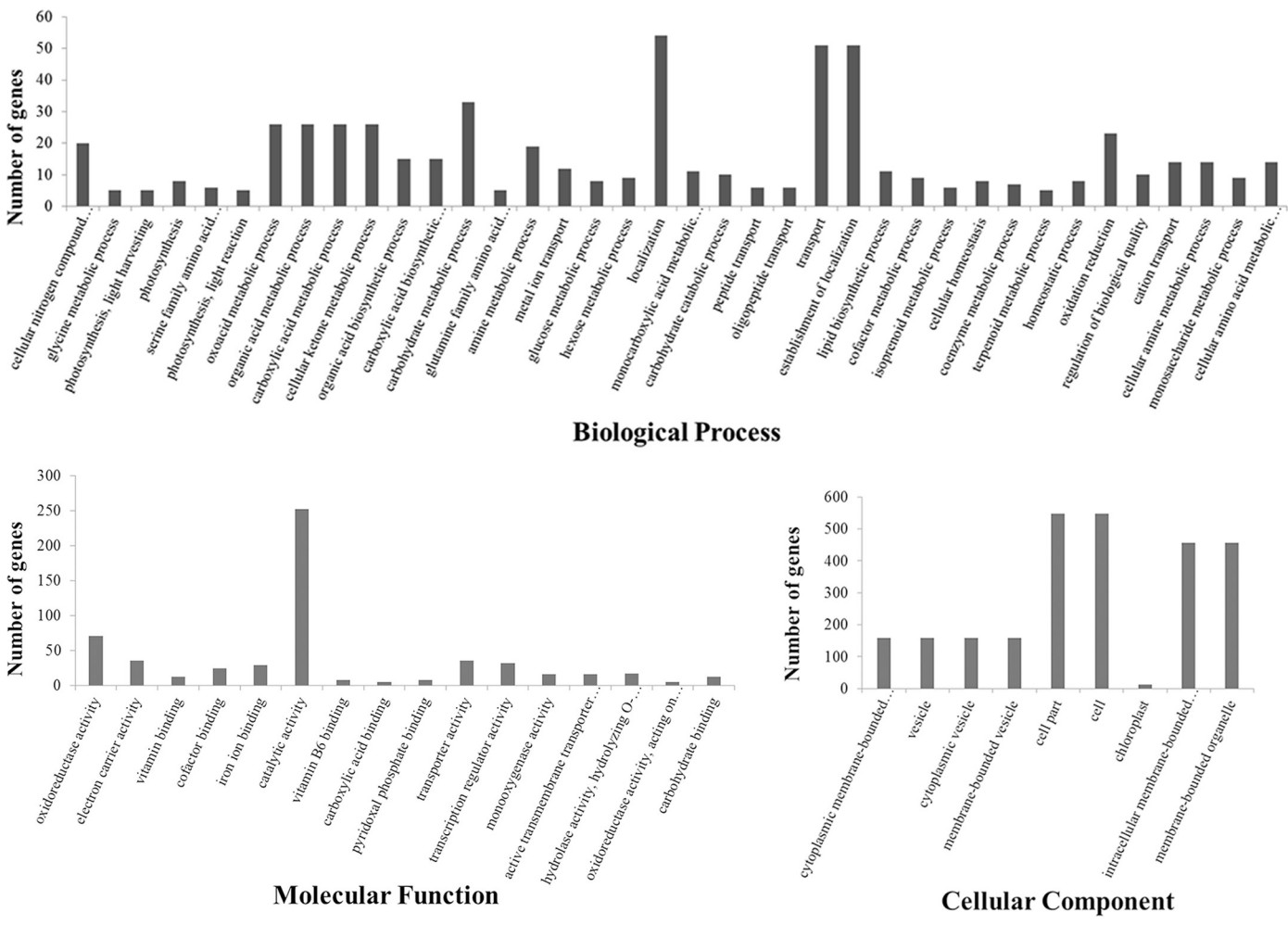

**Fig 3. Significant GO terms of DEGs at the meiosis stage.**

(http://www.gramene.org). A large number of QTLs for yield-related traits were detected, including grain weight per plant, pollen fertility, spikelet number, 1000-grain weight and male fertility restoration. A total of 22 genes were mapped in the interval of 12 yield-related QTLs, including 1 grain number QTL, 1 pollen fertility QTL, 1 spikelet number QTL, 5 1000-grain weight QTLs and 4 male fertility restoration QTLs. Furthermore, 3, 2, 1, 9 and 7 genes were identified in the grain yield per panicle QTLs, pollen fertility QTLs, spikelet number QTLs, 1000-grain weight QTLs and male fertility restoration QTLs, respectively (Table 4). These results showed that several differentially expressed genes were related to yield-related traits in autotetraploid hybrid.

## The mode of inheritance for DEGs$_{HP}$

The 904 and 68 DEGs$_{HP}$ were classified into 12 categories, which were classified into four major expression groups according to the method of Yoo et al. [30], including additive dominance, parental expression level dominance (high parental expression level dominance and low parental expression level dominance), transgressive down-regulation dominance, and transgressive up-regulation dominance. A total of 24, 107, 198, 549, 26 genes were detected in the additive dominance, high parental expression level dominance, low parental expression

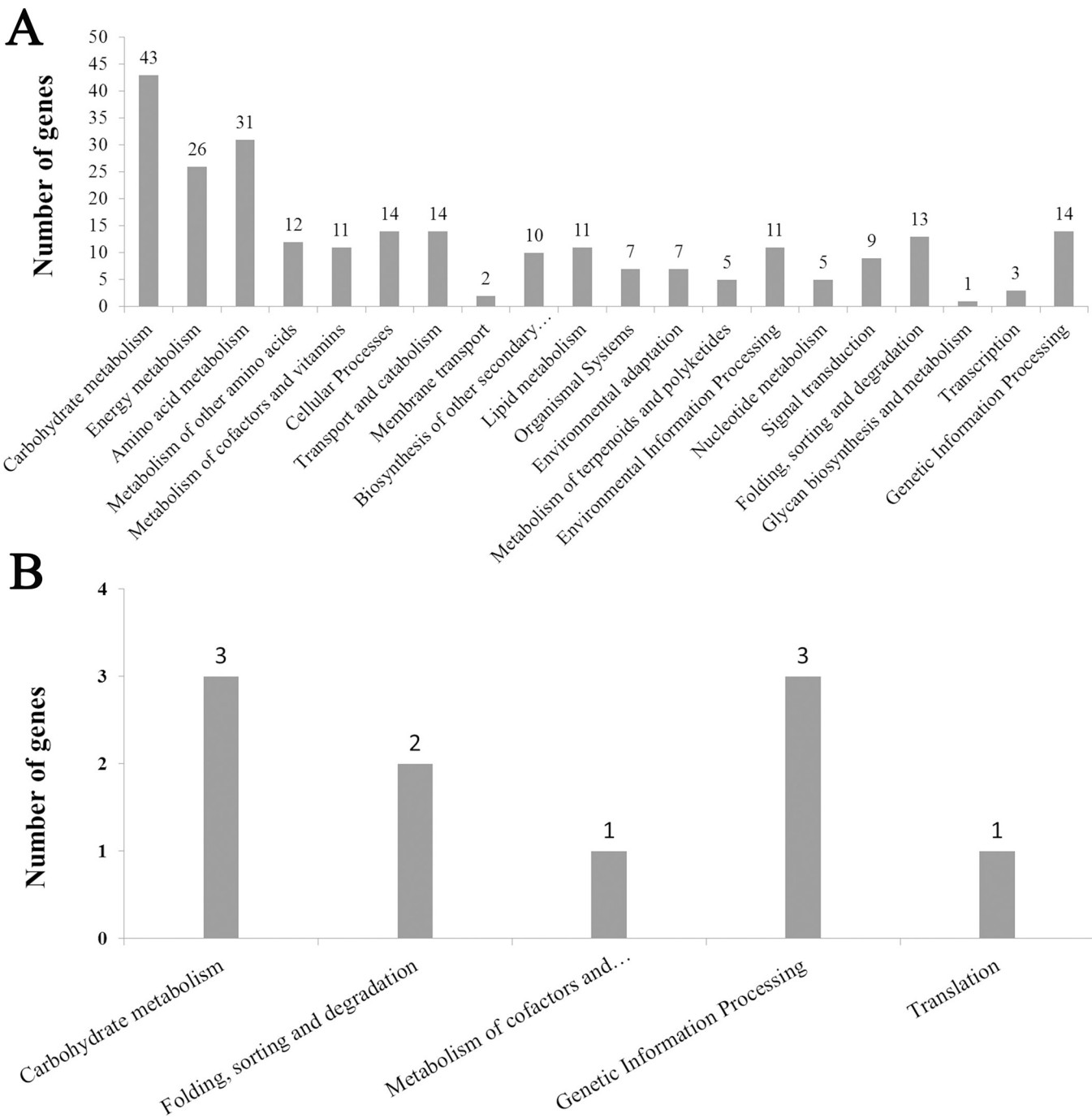

**Fig 4.** KEGG pathways enriched at (A) meiosis and (B) single microspore stage.

level dominance, transgressive down-regulation dominance and transgressive up-regulation dominance at the meiosis stage, and 21, 37, 4, 2 and 4 genes were detected in the additive dominance, high parental expression level dominance, low parental expression level dominance, transgressive down-regulation dominance and transgressive up-regulation dominance at the single microspore stage, respectively (S6 Fig in S1 File). Among the DEGs$_{HP}$, 2.65% and 30.88% had additive dominance, 11.8% and 55.88% had high parental expression level

**Table 3. Significant ko terms of DEGs during meiosis (MA) and single microspore stage (SCP).**

| Term Name | Description | Number of DEGs | p-value |
|---|---|---|---|
| MA | | | |
| ko00630 | Glyoxylate and dicarboxylate metabolism | 17 | 3.26E-11 |
| ko00260 | Glycine, serine and threonine metabolism | 10 | 6.67E-05 |
| ko00280 | Valine, leucine and isoleucine degradation | 7 | 2.25E-04 |
| ko00250 | Alanine, aspartate and glutamate metabolism | 7 | 5.17E-04 |
| ko00195 | Photosynthesis | 7 | 0.001053788 |
| ko00710 | Carbon fixation in photosynthetic organisms | 8 | 0.002301607 |
| ko00910 | Nitrogen metabolism | 5 | 0.003345682 |
| ko00220 | Arginine biosynthesis | 5 | 0.003345682 |
| ko00196 | Photosynthesis—antenna proteins | 3 | 0.008877552 |
| ko00592 | alpha-Linolenic acid metabolism | 5 | 0.009946881 |
| ko04146 | Peroxisome | 7 | 0.015584713 |
| ko00941 | Flavonoid biosynthesis | 4 | 0.017082881 |
| ko00480 | Glutathione metabolism | 7 | 0.018423848 |
| ko00640 | Propanoate metabolism | 3 | 0.022807086 |
| ko00053 | Ascorbate and aldarate metabolism | 4 | 0.03442308 |
| ko00270 | Cysteine and methionine metabolism | 7 | 0.036537999 |
| SCP | | | |
| ko04122 | Sulfur relay system | 1 | 0.020984252 |
| ko00790 | Folate biosynthesis | 1 | 0.038669226 |

dominance, 21.9% and 5.88% had low parental expression level dominance, 60.73% and 2.94% had transgressive down-regulation dominance, and 2.88% and 5.88% had transgressive up-regulation dominance at the meiosis and single microspore stage, respectively (Fig 5).

The higher-parent dominance and transgressive up-regulation dominance were considered autotetraploid potential heterosis genes to investigate heterosis related genes in autotetraploid rice hybrids. A total of 133 and 41 DEGs$_{HP}$ were detected in higher-parent dominance and transgressive up-regulation dominance at the meiosis and single microspore stage, respectively. Among these genes, six genes were found to be commonly expressed in both phases, including one gene encoded beta-amylase (*Os03g0141200*), two new genes (*Oryza_newgene_118* and *Oryza_newgene_309*) and three genes encoding expressed protein (*Os01g0521200, Os01g0612350*

**Table 4. Significant DEGs$_{HP}$ mapped in each of the QTL regions.**

| Trait name | Chr | DEGs$_{HP}$ |
|---|---|---|
| Grain yield per panicle | 5 | *Os05g0438500, Os05g0475400, Os05g0460000* |
| Pollen fertility | 3 | *Os03g0773800, Os03g0786100* |
| Spikelet number | 5 | *Os05g0223000* |
| 1000-grain weight | 6 | *Os06g0320500* |
| 1000-grain weight | 1 | *Os01g0860400, Os01g0869800* |
| 1000-grain weight | 3 | *Os03g0819600, Os03g0844700, Os03g0860100* |
| 1000-grain weight | 7 | *Os07g0170100, Os07g0152800* |
| 1000-grain weight | 7 | *Os07g0624900* |
| Male fertility restoration | 1 | *Os01g0183400, Os01g0151200, Os01g0127900* |
| Male fertility restoration | 11 | *Os11g0210600, Os11g0187500* |
| Male fertility restoration | 10 | *Os10g0356000* |
| Male fertility restoration | 7 | Os07g0624900 |

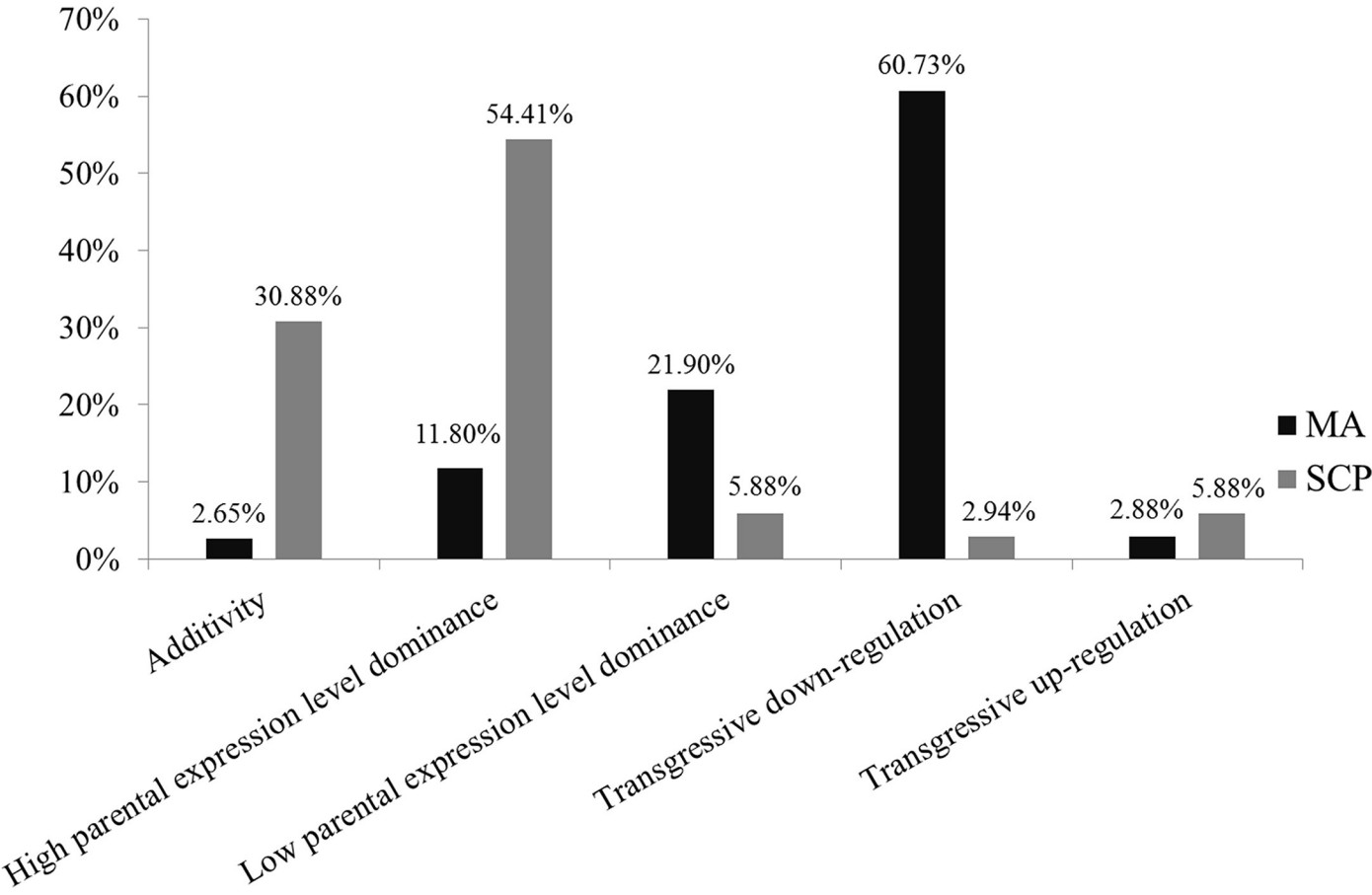

**Fig 5. Classification of the DEGs$_{HP}$ at meiosis and single microspore stages according to the gene expression dominance.**

and *Os05g0136400*). We performed predicted protein-protein interaction analysis at the meiosis stage to detect interactions among these 133 genes, and three sub-networks were detected (Fig 6).

We compared the DEGs$_{HP}$ with the transcriptome data of *Arabidopsis* or rice meiosis stage-specific genes and meiosis-related genes [31–35], and detected one meiosis-related gene (*Os01g0917500*, *MSP1*) and two meiosis specific-genes (*Os07g0624900* and *Os04g0208600*). Besides, one GA-stimulated transcript gene and three transcription factors were detected, including *Os06g0266800*, *Os04g0574500* (GRF), *Os03g0759700* (bHLH), and *Os10g0531900* (bZIP).

## Discussion

### The effect of double neutral genes is stronger in autotetraploid hybrid than diploid hybrid

Autotetraploid rice hybrids between different subspecies (*japonica* and *indica*) have greater adaptability and higher yield potential than the diploid rice hybrid [13–15, 36, 37]. However, pollen abortion caused by multiple genes interaction, i.e. *Sa*, *Sb* and *Sc* pollen sterility loci, is vital factor that causes abortion of male meiocytes and low pollen fertility in autotetraploid and diploid rice hybrids. Polyploidy increased the multi-allelic interaction at three loci that enhanced chromosomal abnormalities compared to diploid rice [17, 18, 20, 25]. It had been

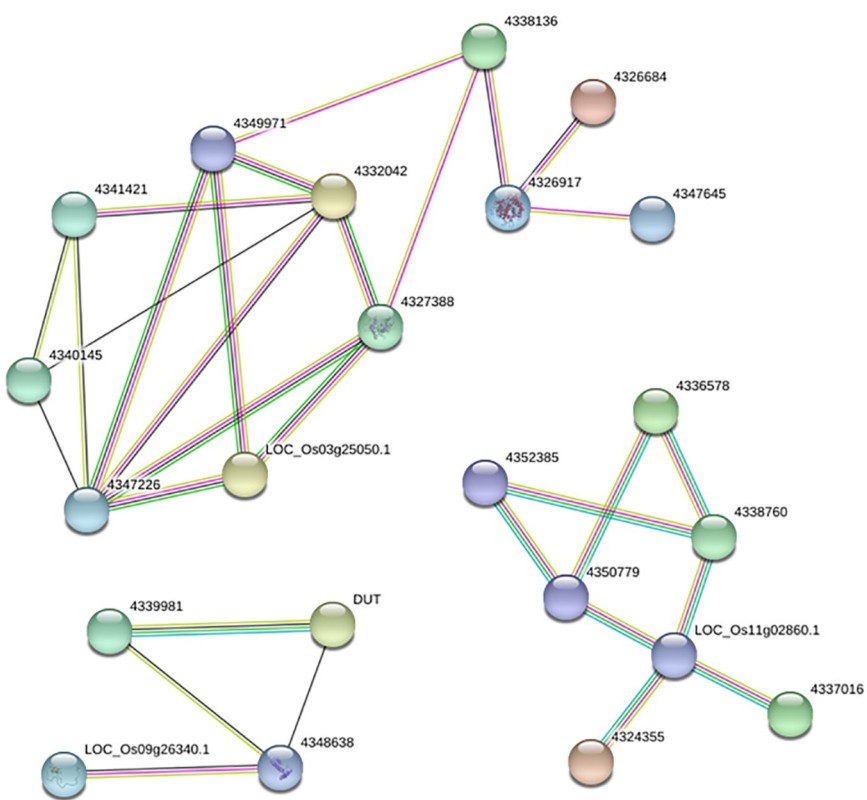

**Fig 6. Predicted protein-protein interaction networks of 133 DEGs were detected in higher-parent dominance and transgressive up-regulation dominance at the meiosis.**

demonstrated that intersubspecific autotetraploid and diploid hybrid rice sterility could be overwhelmed by double neutral genes ($S_a^n$ and $S_b^n$) [7, 20]. All autotetraploid rice hybrids harboring these double neutral genes ($S_a^n$ and $S_b^n$) showed normal pollen fertility (>70%), some of them generated by crossing of neo-tetraploid rice with autotetraploid even exhibited high seed set and significant positive heterosis for yield and yield-related traits [22, 24]. Interestingly, the results of the present study indicated that the pollen fertility of autotetraploid hybrid with double neutral genes ($S_a^n$ and $S_b^n$) was similar to the parents. In contrast, pollen fertility in diploid hybrid was lower than its parents. These results were consistent with our previous study [25], who also detected high fertility in autotetraploid rice hybrids.

Analysis of the agronomic traits in both autotetraploid and diploid hybrids showed that the values for HPH and MPH of autotetraploid hybrid were positive for all the traits except grain length and grain width. In particular, the values of MPH were very high for FG and SS, which were 149.22% and 214.68%, respectively. The autotetraploid hybrid also showed positive and the highest HPH for these two traits, and the HPH values were 62.21% and 158.33%, respectively. However, MPH and HPH values for FG were 41.98% and -3.96%, and 5.61% and -4.35% for SS in diploid hybrid, respectively. These results indicated that autotetraploid hybrid displayed higher heterosis than diploid hybrid. Similarly, high heterosis values were detected in autotetraploid rice compared to diploid rice [14, 15, 37].

It is well known that meiosis is a critical process, and chromosome behavior had an essential impact on pollen fertility and seed setting in rice. Recent research showed that the frequency of normal chromosome behavior was higher in hybrids harboring $S_a^n$ and $S_b^n$ neutral genes than their parents [22, 24]. Here, the chromosomal abnormalities of autotetraploid

hybrid with double neutral genes was lower than its parents in metaphase I, metaphase II, anaphase I and anaphase II. But the frequency of chromosome behavior of the diploid rice hybrids carrying $S_a{}^n$ and $S_b{}^n$ neutral genes and its parents were similar. All these results demonstrated that neutral genes had a more significant impact on chromosome behavior in autotetraploid hybrids than diploid hybrids. Hence, we inferred that double neutral genes have stronger effect on autotetraploid rice than diploid rice.

## Possible reasons for gene expression differences in different ploidy rice hybrids

Meiosis plays a crucial role in rice pollen development. Transcriptome analysis was most often used to explore the effect of interactions between pollen sterility loci and polyploidy on gene expression profiles in autotetraploid rice hybrids during PMC meiosis. In total, 55 meiosis stage-specific or meiosis-related genes increased pollen sterility loci interactions in autotetraploid rice hybrids [20]. The 26 meiosis-stage specific and four meiosis-related genes displayed up-regulation in autotetraploid hybrid harboring $S_a{}^n$ and $S_b{}^n$ neutral genes and paternal line compared to maternal line [24]. Here, 904 and 68 DEGs$_{HP}$ were specifically expressed in autotetraploid hybrid at meiosis and single microspore stages compared with diploid hybrid. The higher-parent dominance and transgressive up-regulation dominance genes were considered as autotetraploid potential heterosis genes, and 133 and 41 such genes were detected at the meiosis and single microspore stage, respectively. A meiosis-related gene (*Os01g0917500*, *MSP1*) and two meiosis-specific genes (*Os07g0624900* and *Os04g0208600*) were identified. *MSP1* gene encodes a Leu-rich repeat receptor-like protein kinase and is required to initiate anther wall formation in rice and to restrict the number of cells entering into female and male sporogenesis [38]. The meiosis specific-gene (*Os07g0624900*) encoded SKP1-like protein that is necessary for multiple cellular processes in eukaryotes, such as ethylene, jasmonate, gibberellin (GA), auxin and light responses [39]. *ASK1* contributes to the regulation of synapsis and synaptonemal complex formation during early meiotic prophase, homolog juxtaposition, chromosome remodeling in Arabidopsis [40, 41]. *LSK1*, *LSK2*, *LSK3* were specifically expressed in late pollen development stages and the elongating pollen tube in three Lily [42]. Another meiosis specific-gene (*Os04g0208600*) encoded F-box/FBD protein. The *S*-locus F-box protein is recognized as the pollen determining factor of S-RNase-based self-incompatibility in Rosaceae, Scrophulariaceae and Solanaceae [43]. In crops, some F-box genes are associated with flower development and flowering [44]. These results suggested that the expression profiles of these meiosis-specific or meiosis-related genes regulate the fertility of autotetraploid rice hybrid harboring double neutral genes.

Transcription factors (TFs) play vital roles in gene regulatory networks, and the interactions between TFs and their target genes regulate spatiotemporal gene expression levels. In the present research, three transcription factors were found to be encoded by the identified DEGs, including *Os04g0574500* (GRF), *Os03g0759700* (bHLH) and *Os10g0531900* (bZIP). The GRF family plays a vital role in cell division and proliferation during leaf development in rice [45]. *OsGRF12* (*Os04g0574500*) encodes a putative novel transcriptional regulator in rice, and its expression enhanced after treating with GA$_3$ [46]. BHLH and bZIP transcription factors have been reported to involved in floral development, such as the pollen development and pollen fertility and floral transition and initiation [25, 47]. Signalling pathways, controlling many cellular processes, were mostly affected by TFs. However, there is little known about the genes regulated by TFs and their particular roles in rice plant metabolism. There seems to be several genes involved in the Leu-rich repeat receptor-like protein kinase gene, hormone-related genes (auxin transporter, gibberellin-regulated protein), F-box protein gene, GRF, bHLH,

bZIP transfactor family. These genes take part in many biological pathways, including cellular processes, transcription factors, signal transduction mechanisms and photosynthesis. These results would offer novel insights into the interaction network associated with male fertility in autotetraploid rice hybrid.

## The role of meiosis, photosynthesis and heterosis related genes in rice harboring double neutral genes

Although heterosis has been widely investigated in rice breeding and plays a vital role in agriculture, our understanding of the molecular mechanism involved in heterosis is still weak. RNA-seq provides a very convenient platform to investigate the molecular basis of heterosis in rice. The high-throughput gene-expression profiling in heterotic cross combinations has been implemented to find a large number of DEG between hybrids and their parents [48]. Here, we revealed various genes associated with strong heterosis in autotetraploid hybrid by transcriptional profiling. For example, *OsAMTR1* (*Os05g0475400*), which is related to grain yield per panicle, encodes aminotransferase that have different isoforms expressing in mitochondria, cytosol and peroxisomes and associated with different cellular processes [49, 50]. *GLO1* (*Os03g0786100*), which is related to pollen fertility, encodes a protein with 369 amino acids, and is predominately expressed in rice leaves [51]. Glycolate oxidase (GLO) is a key enzyme in photorespiratory metabolism, and *PsbS1* (*Os01g0869800*) is related to 1000-grain weight, which encodes photosystem protein. The accumulation of this protein exerts control over photosynthesis in fluctuating light [52].

Functional analysis of hybrid transcriptomes indicated that these genes were involved in multiple metabolisms, which play crucial roles in meiosis and heterosis and participate in most vital metabolic pathways, such as amino acid metabolism, photosynthesis metabolism. Amino acid metabolism not only provides vital raw resources for synthesizing proteins, polypeptides and other nitrogen-containing substances but also provides the materials for the maintenance of life and normal metabolism. In this study, the analyzed data showed that 31 DEGs involved in the different amino acid metabolism were up-regulated at the meiosis. Many recent studies demonstrated that the balance of free amino acids are vital for the pollen fertility and anther development, and the concentrations of certain amino acids in maize leaves are strictly related to hybrid yield [53, 54].

The amino acids metabolism involved in this study included leucine, valine and isoleucine degradation, serine, glycine and threonine metabolism, aspartate, alanine and glutamate metabolism, cysteine and methionine metabolism. Serine metabolism is mostly related to photorespiration in plants where two molecules of glycine constitute and two other pathways of serine synthesis which represent the branches of glycolysis [55–57]. The glycerate and phosphorylated pathways of serine synthesis in plants are the important process linking carbon and nitrogen metabolism [58]. Complete oxidation of Valine, Leucine and Isoleucine effectively allows the formation of ATP in the mitochondria by oxidative phosphorylation. But it is mostly unknown about the metabolic pathways for these branched-chain amino acids breakdown so far in plants [59]. Glutamate is an active amino acid, which rapidly stimulated the expression of genes involved in signal transduction growth and transport [60]. Methionine, is a building block for protein synthesis, is the immediate precursor of S-adenosylmethionine which plays numerous roles in transmethylation reactions and the biosynthesis of polyamines and the phytohormone ethylene [61, 62].

Photosynthesis is an important biological process on this planet, which provides consumable energy for plant development. Heterosis is associated with increased photosynthesis. The photosynthesis process can be divided into biochemical and fluorescence processes, which is a

primary process converting $CO_2$ into organic compounds through solar energy [63]. Here, the $DEGs_{HP}$ was significantly enriched in the photosynthesis metabolism process between autotetraploid hybrid and its parents. We found that 8 DEGs were involved in carbon fixation of photosynthesis (ko00710), including genes encoding photosystem II, ATPase. 7 and 3 DEGs were involved in photosynthesis (ko00195) and photosynthesis–antenna proteins (ko00196), including gene encoding chlorophyll A-B binding protein. GO annotation of cellular component showed that the DEGs significantly enriched in photosynthesis-related organelles. These results showed that the metabolic pathways of amino acids and photosynthesis have a more significant influence on the yield of autotetraploid rice hybrid compared to other metabolic pathways. The genes within yield QTLs involved in these two metabolic pathways are important candidate genes for heterosis and yield and probably associated with the high yield and heterosis in autotetraploid rice hybrid.

## Supporting information

**S1 File.** S1 Fig Chromosome behaviors during PMC meiosis in $DF_1$ (diploid hybrid); S2 Fig Chromosome behaviors during PMC meiosis in $AF_1$ (autotetraploid hybrid); S3 Fig Morphological characteristics of $F_1$ hybrid and their parents with different ploidy levels. S4 Fig The correlation coefficient of different ploidy hybrids and their parents in anthers at meiosis and single microspore stages. S5 Fig Confirmation of the DEGs in T449 and autotetraploid $F_1$ hybrid during meiosis stage. S6 Fig Twelve possible additive and nonadditive gene expression patterns in autotetraploid hybrid relative to its parents.
(PDF)

**S2 File.** S1 Table. List of primers used for qRT-PCR. S2 Table. Number of observed cells and frequency of normal cells in different ploidy hybrids and parents. S3 Table Heterosis analysis of autotetraploid and diploid rice parents and $F_1$ hybrids. S4 Table The data of different ploidy hybrids and their parents in anthers at meiotic and single microspore stages.
(XLSX)

## Acknowledgments

The authors thank Ms. Shuhong Yu and other lab members for assistance.

## Author Contributions

**Conceptualization:** Lin Chen, Haibin Guo, Muhammad Qasim Shahid.

**Data curation:** Muhammad Qasim Shahid.

**Formal analysis:** Lin Chen, Haibin Guo, Shuling Chen, Huijing Yang, Fozia Ghouri, Muhammad Qasim Shahid.

**Funding acquisition:** Haibin Guo, Muhammad Qasim Shahid.

**Investigation:** Lin Chen, Haibin Guo, Shuling Chen.

**Methodology:** Lin Chen, Haibin Guo, Shuling Chen, Huijing Yang, Fozia Ghouri.

**Project administration:** Muhammad Qasim Shahid.

**Resources:** Muhammad Qasim Shahid.

**Supervision:** Muhammad Qasim Shahid.

**Validation:** Lin Chen, Haibin Guo, Huijing Yang, Fozia Ghouri.

**Writing – original draft:** Lin Chen, Haibin Guo, Shuling Chen, Huijing Yang, Fozia Ghouri, Muhammad Qasim Shahid.

**Writing – review & editing:** Lin Chen, Haibin Guo, Fozia Ghouri, Muhammad Qasim Shahid.

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
