## [Decision Letter · Decision Letter 0]

1 Jul 2020

PONE-D-20-14708

Comparative study on cytogenetics and transcriptome between diploid and autotetraploid rice hybrids harboring double neutral genes

PLOS ONE

Dear Dr. Shahid,

Thank you for submitting your manuscript to PLOS ONE. After careful consideration, we feel that it has merit but does not fully meet PLOS ONE’s publication criteria as it currently stands. Therefore, we invite you to submit a revised version of the manuscript that addresses the points raised during the review process.

I have now received comments on your papers from reviewers. You will see that they are advising major revision of your manuscript. Therefore you can request to add the query raised by reviewers and to submit the revised version of your manuscript

We look forward to receiving your revised manuscript.

Kind regards,

Allah Bakhsh

Academic Editor

PLOS ONE

Journal Requirements:

Reviewers' comments:

Reviewer's Responses to Questions

**Comments to the Author**

1. Is the manuscript technically sound, and do the data support the conclusions?

Reviewer #1: Yes

Reviewer #2: Yes

Reviewer #3: Yes

2. Has the statistical analysis been performed appropriately and rigorously? 

Reviewer #1: Yes

Reviewer #2: Yes

Reviewer #3: Yes

3. Have the authors made all data underlying the findings in their manuscript fully available?

Reviewer #1: Yes

Reviewer #2: Yes

Reviewer #3: Yes

4. Is the manuscript presented in an intelligible fashion and written in standard English?

Reviewer #1: Yes

Reviewer #2: Yes

Reviewer #3: Yes

5. Review Comments to the Author

Reviewer #1: In this paper Chen et al., 2020, compared cytological and transcriptome data between autotetraploid and diploid rice hybrid during meiosis and single microspore stages to understand the molecular mechanism of neutral genes for overcoming pollen sterility in autotetraploid rice hybrids. Using cytological analysis they identified lower chromosomal abnormalities in autotetraploid compared to other lines. The authors provide a theoretical basis for molecular mechanisms of heterosis in autotetraploid rice.

I would suggest to improve the quality of the figures and have some minor comments for the paper:

- Page 5, Line 2-3, which lines were crossed?

- Page 8, in the first paragraph, authors needs to describe the general design of experiment, which lines they crossed and which diploid and tetraploid lines they developed for analysis. Also authors needs to explain what they are showing in the figure.

- The nomenclature of hybrids used in Figure 2 is different than what is described at Page 8, line 19-21

- Page 9, Line 19-20, authors needs to describe at least some of the genes validated by qRT-PCR

- Page 10, Line 20, please explain what these results show?

- Similarly Page 12, Line 17, what you want to show by these results?

Reviewer #2: The manuscript entitled “Comparative study on cytogenetics and transcriptome between diploid and autotetraploid rice hybrids harboring double neutral genes” is focused on the role of neutral genes in diploid and autotetraploid rice. The mechanism of pollen sterility is very complex and hot issues for rice breeders. Neutral genes are great source to overcome reproductive isolation in indica-japonica hybrids. The results indicated that meiosis-specific and meiosis-related genes, and amino acids and photosynthesis metabolism-related genes contribute to higher yield and pollen fertility in autotetraploid rice hybrid. Overall, the manuscript is well written and executed but there are few issues which should be resolved before final acceptance.

Major Concern

Authors have published an article about double neural genes previously (Wu et al. 2017; Rice). What are major differences between these studies. If any data similar, should cite a reference. How about data submission to NCBI?

Minor concerns

P7L21, P8L1-5, please delete/re-write from the results section.

P8 Authors could add more cytological results, such as images about abnormal chromosome behavior in diploid and autotetraploid rice.

Figure 1 legends need to improve.

Reviewer #3: The authors try to clarify the role of double neutral genes associated with pollen fertility in autotetraploid rice hybrid; however, all results are on the way to the goal and far from the conclusions that the authors claimed in the Discussion. There are several mistakes in the manuscript that authors need to properly addressed. I have following concerns about the manuscript:

Introduction

Page 5 line 2-4: ‘we developed autotetraploid and diploid rice hybrids by crossing T449, …’ Are autotetraploid and diploid rice all crossing T449? I think this is a wrong sentence.

Methods

Page 6 line 22. gene Ontology should be Gene Ontology

Results

Page 8 line 1-7: This paragraph is almost the introduction.

Page 8 line 8-9: Please re-phrase.

Page 8 line 11: I cannot see the difference in pollen fertility between the materials in figure 1 clearly. The seed set of the materials should be present in the manuscript.

Page 8 line 15-21: How many PMCs at each stage during meiosis the authors observed? Why are there no figures of the abnormal chromosome behavior in DF1 and TF1?

Page 8 line 19: “…was lower than its parents in the metaphase I, …” I could only found T449 in the figure 2, where are the data of T431, T435 and T438 during meiosis?

Page 9 line 2: Lower pollen fertility of diploid hybrid (~70%) was found, the seed set of diploid hybrid was similar to the parents (90%). The pollen fertility of TF1 was similar to DF1, but the seed set is lower than diploid hybrid. How to explain it? What’s the connection of Double neutral genes in TF1 and DF1? What’s the purpose of this analysis? To evaluate the heterosis of autotetraploid hybrid and diploid hybrid, or ploidy analysis between TF1 and DF1, or double neutral genes analysis, it makes me very confusing.

Page 9 line 17: the table S4 should be present in the figure.

Page 9 line 18: Figure S4, the authors randomly selected 10 genes for qPCR assay, why all these genes showed up-regulation. How about the down-regulated genes?

Discussion

The discussion is not so well written. How to get the conclusion of the first subhead in the discussion? Is this only base on hybrid vigour? “But the frequency of chromosome behavior of the diploid rice hybrids carrying San and Sbn neutral genes and its parents were similar” How to get this result? Where are the statistical results of this part? I CAN’T FIND IN THE RESULTS SECTION.

6. PLOS authors have the option to publish the peer review history of their article (what does this mean?). If published, this will include your full peer review and any attached files.

Reviewer #1: No

Reviewer #2: No

Reviewer #3: No

---

## [Author Response · Author response to Decision Letter 0]

11 Aug 2020

Please see attached file "Response to Reviewers" for detail. Thanks

---

## [Decision Letter · Decision Letter 1]

28 Aug 2020

PONE-D-20-14708R1

Comparative study on cytogenetics and transcriptome between diploid and autotetraploid rice hybrids harboring double neutral genes

PLOS ONE

Dear Dr. Shahid,

Thank you for submitting your manuscript to PLOS ONE. After careful consideration, we feel that it has merit but does not fully meet PLOS ONE’s publication criteria as it currently stands. Therefore, we invite you to submit a revised version of the manuscript that addresses the points raised during the review process.

I have received comments on your revised version of your manuscript. You are kindly requested to address some of minor comments by one of the reviewers. Overall the article seems much improved. You can kindly requested to return the manuscript within a week after the minor changes are adhered to it.

We look forward to receiving your revised manuscript.

Kind regards,

Allah Bakhsh

Academic Editor

PLOS ONE

Reviewers' comments:

Reviewer's Responses to Questions

**Comments to the Author**

1. If the authors have adequately addressed your comments raised in a previous round of review and you feel that this manuscript is now acceptable for publication, you may indicate that here to bypass the “Comments to the Author” section, enter your conflict of interest statement in the “Confidential to Editor” section, and submit your "Accept" recommendation.

Reviewer #1: (No Response)

Reviewer #2: All comments have been addressed

Reviewer #3: All comments have been addressed

2. Is the manuscript technically sound, and do the data support the conclusions?

Reviewer #1: (No Response)

Reviewer #2: Yes

Reviewer #3: Yes

3. Has the statistical analysis been performed appropriately and rigorously? 

Reviewer #1: (No Response)

Reviewer #2: Yes

Reviewer #3: Yes

4. Have the authors made all data underlying the findings in their manuscript fully available?

Reviewer #1: (No Response)

Reviewer #2: Yes

Reviewer #3: Yes

5. Is the manuscript presented in an intelligible fashion and written in standard English?

Reviewer #1: (No Response)

Reviewer #2: Yes

Reviewer #3: Yes

6. Review Comments to the Author

Reviewer #1: (No Response)

Reviewer #2: (No Response)

Reviewer #3: The authors have revised the manuscript carefully according to reviewers’ suggestion. The authors have added new table and figures, revised sentences in the results section, and figures and figure legends. So, readers can follow the contents easily and obtain very useful information from the paper. This manuscript could be accepted for publication after being incorporated few minor suggestions as follow.

1. In Plant material: It is mentioned that “E249, and its autotetraploid rice line, T449, harboring San and Sbn”. The authors need to write the origin of these materials in detail.

2. Page 27. Bar=100um maybe Bar=100µm.

3. Please cross check all the references carefully.

4. Please be consistent with line name, E24-2x could be change as E24

7. PLOS authors have the option to publish the peer review history of their article (what does this mean?). If published, this will include your full peer review and any attached files.

Reviewer #1: No

Reviewer #2: No

Reviewer #3: No

---

## [Author Response · Author response to Decision Letter 1]

1 Sep 2020

Dear Editor

Thank you very much for taking interest in our research article (PONE-D-20-14708R1). We are truly grateful to yours and other reviewers’ critical comments and valuable suggestions. We have revised the manuscript according to reviewers' suggestions and have incorporated all the changes suggested by reviewer. We acknowledge the reviewers for their valuable suggestions that considerably improved the quality and contents of the manuscript. Following is our point-by-point response to the reviewer’s comments:

Reviewers' comments:

Reviewer's Responses to Questions

Comments to the Author

1. If the authors have adequately addressed your comments raised in a previous round of review and you feel that this manuscript is now acceptable for publication, you may indicate that here to bypass the “Comments to the Author” section, enter your conflict of interest statement in the “Confidential to Editor” section, and submit your "Accept" recommendation.

Reviewer #1: (No Response)

Reviewer #2: All comments have been addressed

Reviewer #3: All comments have been addressed

Response: We are very thankful to reviewers for encouraging comments about our manuscript.

2. Is the manuscript technically sound, and do the data support the conclusions?

Reviewer #1: (No Response)

Reviewer #2: Yes

Reviewer #3: Yes

Response: The reviewers agreed with the quality of manuscript and we have further revised the manuscript carefully based on the reviewer suggestions. 

3. Has the statistical analysis been performed appropriately and rigorously?

Reviewer #1: (No Response)

Reviewer #2: Yes

Reviewer #3: Yes

Response: The reviewers agreed with the statistical analysis performed in the manuscript. Thanks

4. Have the authors made all data underlying the findings in their manuscript fully available?

Reviewer #1: (No Response)

Reviewer #2: Yes

Reviewer #3: Yes

Response: The reviewers are agreed that manuscript adhere to the PLOS Data Policy. 

5. Is the manuscript presented in an intelligible fashion and written in standard English?

Reviewer #1: (No Response)

Reviewer #2: Yes

Reviewer #3: Yes

Response: The reviewers are agreed that manuscript is presented in an intelligible fashion and written in standard English. 

6. Review Comments to the Author

Reviewer #1: (No Response)

Reviewer #2: (No Response)

Reviewer #3: The authors have revised the manuscript carefully according to reviewers’ suggestion. The authors have added new table and figures, revised sentences in the results section, and figures and figure legends. So, readers can follow the contents easily and obtain very useful information from the paper. This manuscript could be accepted for publication after being incorporated few minor suggestions as follow.

Response: We are very thankful to reviewer for encouraging comments about our manuscript and we have tried our best to improve the manuscript as suggested by reviewer.

1. In Plant material: It is mentioned that “E249, and its autotetraploid rice line, T449, harboring San and Sbn”. The authors need to write the origin of these materials in detail.

Response: We have mentioned the origin of these cultivars in M&M section. Thanks

2. Page 27. Bar=100um maybe Bar=100µm.

Response: Thanks for indicaitng a mistake in figure legends, and we have revised it according to reviewer suggestion.

3. Please cross check all the references carefully.

Response: We have carefully revsied references in the manuscript.

4. Please be consistent with line name, E24-2x could be change as E24

Response: Thanks for useful suggestion and we have mentioned it as E24 in whole manuscript.

Again, we appreciate all of your insightful comments. Thank you for taking your time and energy to help us to improve the manuscript. We hope that the revised version of our manuscript is now suitable for publication in PLOSONE.

Sincerely,

MQ Shahid (on behalf of all co-authors)

Associate Professor

College of Agriculture

South China Agricultural University

Guangzhou, 510642, P.R. China

---

## [Editor Report · Decision Letter 2]

7 Sep 2020

Comparative study on cytogenetics and transcriptome between diploid and autotetraploid rice hybrids harboring double neutral genes

PONE-D-20-14708R2

Dear Dr. Shahid,

We’re pleased to inform you that your manuscript has been judged scientifically suitable for publication and will be formally accepted for publication once it meets all outstanding technical requirements.

Kind regards,

Allah Bakhsh

Academic Editor

PLOS ONE
---

## [Editor Report · Acceptance letter]

17 Sep 2020

PONE-D-20-14708R2 

Comparative study on cytogenetics and transcriptome between diploid and autotetraploid rice hybrids harboring double neutral genes 

Dear Dr. Shahid:

I'm pleased to inform you that your manuscript has been deemed suitable for publication in PLOS ONE. Congratulations! Your manuscript is now with our production department. 

Kind regards, 

on behalf of

Dr. Allah Bakhsh 

Academic Editor

PLOS ONE